# Deep Learning-Based Joint CSI Feedback and Hybrid Precoding in FDD mmWave Massive MIMO Systems

**DOI:** 10.3390/e24040441

**Published:** 2022-03-23

**Authors:** Qiang Sun, Huan Zhao, Jue Wang, Wei Chen

**Affiliations:** 1School of Information Science and Technology, Nantong University, Nantong 226019, China; sunqiang@ntu.edu.cn (Q.S.); 2010310019@stmail.ntu.edu.cn (H.Z.); chenwei0303@ntu.edu.cn (W.C.); 2Nantong Research Institute for Advanced Communication Technologies (NRIACT), Nantong 226019, China

**Keywords:** deep learning, massive MIMO, CSI feedback, hybrid precoding, millimeter wave

## Abstract

In this paper, we propose an end-to-end deep learning approach to realize channel state information (CSI) feedback and hybrid precoding for millimeter wave massive multiple-input multiple-output systems in the frequency division duplexing mode. Different from conventional approaches that treat the CSI reconstruction and hybrid precoding as separate components, we propose a new end-to-end learning method bypassing the channel reconstruction phase, and design the hybrid precoders and combiners directly from the feedback codewords (a compressed version of the CSI). More specifically, we design a neural network composed of the CSI feedback and hybrid precoding. Experiment results show that our proposed network can achieve better performance than conventional hybrid precoding schemes that reserve channel reconstruction, especially when the feedback resources are limited.

## 1. Introduction

Hybrid precoding is a promising technique for millimeter wave (mmWave) massive multiple-input multiple-output (MIMO) systems [1,2,3,4,5], thanks to its merit of reducing the number of RF chains while achieving a similar performance of fully digital architecture [1]. Many algorithms for hybrid precoding design have been proposed [2,3,6,7,8,9,10], e.g., the simultaneous orthogonal matching pursuit (SOMP) algorithm [2] and the manifold optimization based alternate minimization (MO-AltMin) algorithm [6]. In these works, it is critical for the base station (BS) to acquire accurate downlink channel state information (CSI) [11]. In frequency division duplexing (FDD) communication systems, however, it is challenging for the BS to acquire the downlink CSI since the uplink-downlink channel reciprocity does not hold. Hence, the user equipments (UEs) need to estimate the downlink CSI and report it to the BS through feedback links. Conventional CSI feedback schemes utilize techniques such as codebook design [12,13] and compressive sensing (CS) [14]. There are also some works that jointly optimize CSI feedback and hybrid precoding [15,16]. The authors in [15] proposed a two-stage approach based on long-term and instantaneous CSI to realize hybrid precoding design and reduce feedback overhead for FDD multiuser massive MIMO systems. The paper [16] investigated the performance of hybrid precoding based on quantized CSI feedback for multi-user massive MIMO systems. However, these approaches introduce an additional implementation cost and overhead to the system, especially when the number of users and the number of BS antennas are large.

In recent years, with the rapid development of deep learning technologies, many new approaches have been realized successfully in CSI feedback [11,17,18,19,20,21] and hybrid precoding [22,23,24,25]. As for the CSI feedback, a deep learning-based CSI compression and recovery scheme, named CsiNet, has been proposed in [11]. The CSI reconstruction accuracy of the CsiNet significantly outperforms existing CS algorithms [26]. Stemming from the CsiNet, more sophisticated architectures have been further developed to enhance the performance, e.g., CsiNet-LSTM [17], CsiNet+ [18]. The paper [20] further reduced the feedback overhead by feeding back bit streams instead of floating point numbers. Considering the noise in the practical feedback link, the authors in [21] proposed a denoising network to reduce the effect of noise on CsiNet. As for the hybrid precoding, the authors of [22] utilized the multi-layer perceptrons (MLP) to design the precoders. In [23,24], convolutional neural network (CNN) frameworks were proposed to estimate the analog precoders and combiners for the single-user and multi-user scenarios, respectively. The authors in [27] use the quantized received signal strength indicators for hybrid precoding design. In [28], the authors proposed a deep learning framework for hybrid precoding and channel estimation without instantaneous CSI feedback for mmWave massive MIMO systems.

Most of the aforementioned works realize CSI feedback and hybrid precoding in separate modules. However, such a separate design may not fully explore the capabilities and advantages that can be provided by end-to-end deep learning [29,30]. In this regard, several recent works focus on exploring end-to-end deep learning that bypasses different intermediate components of communication systems [29,30,31,32]. The authors in [29,32] proposed an end-to-end design that bypasses the channel estimation, and designs the hybrid precoders directly from the received pilots in TDD massive MIMO systems. Refs. [30,31] proposed the idea of bypassing channel reconstruction in FDD massive MIMO systems. In [30], considering a point-to-point MIMO system, the authors jointly train the DNNs at the transmitter (TX) and the receiver (RX) where the DNN in the receiver is used to map the pilot-aided signals into quantized vectors, and the DNN in the transmitter is used to map the quantized vectors into precoding vectors. In [31], the authors treats the end-to-end precoding design problem as a distributed source coding (DSC) problem and jointly designs the downlink pilot training, channel feedback, and precoding. However, Refs. [30,31] only consider the design of fully-digital precoding. The end-to-end design of CSI feedback and hybrid precoding that bypasses channel reconstruction in FDD mmWave massive MIMO is less understood in the literature and is still open for investigation.

In this paper, we investigate the joint design of CSI feedback and hybrid precoding for FDD massive MIMO systems. We propose a new neural network structure and an end-to-end learning framework, which bypasses channel reconstruction and directly designs the hybrid precoders and combiners from the feedback codewords. Specifically, our proposed neural network consists of two parts: CSI feedback and hybrid precoding. In order to train the network, we generate the input-output pairs where the input is the channel matrices and the output is the hybrid precoders and combiners. The main contributions of this paper are summarized as follows:A new deep learning-based end-to-end method of joint CSI feedback and hybrid precoding for FDD massive MIMO systems is proposed. Differing from the existing works that jointly optimize CSI feedback and hybrid precoding by using traditional algorithms, we adopt end-to-end deep learning techniques to solve the problem. Meanwhile, our proposed method bypasses channel reconstruction and directly designs the hybrid precoders and combiners from the feedback codewords for FDD massive MIMO systems, which is different from prior works that treat the CSI reconstruction and hybrid precoding as separate components and has been less investigated in the latest end-to-end works;A new end-to-end neural network structure for FDD mmWave massive MIMO systems is proposed in this paper. It consists of two parts: CSI feedback and hybrid precoding. The former, realized by CNN, transforms the channel matrices into feedback codewords and the latter, realized by DNN, transforms feedback codewords into hybrid precoders and combiners;The simulation results illustrate that compared with conventional approaches, which reserve channel reconstruction, our proposed method can significantly reduce the feedback overhead and achieve better performance, especially when the feedback resources are limited.

**Notation** **1.**
*We use ${\left(\cdot \right)}^{T}$, ${\left(\cdot \right)}^{H}$, ${\left(\cdot \right)}^{-1}$, which denote transpose, conjugate transpose and inverse, respectively. $I_{N}$ denotes the identity matrix whose size is N×N. AB represents the number of all combinations of B elements taken from A different elements. $ℜ\left\{\cdot \right\}$ and $ℑ\left\{\cdot \right\}$ denote the real and imaginary parts of a variable. $∠\left\{\cdot \right\}$ denotes the angle of complex quantity. $E\left\{\cdot \right\}$ denotes the statistical expectation. ${\left[Y\right]}_{i,j}$ denotes (i-th, j-th) element of matrix Y. ${\left[Y\right]}_{:,j}$ denotes j-th column of matrix Y. $\left|Y\right|$ denotes the determinant of matrix Y.*


## 2. System Model

We consider a point-to-point FDD mmWave MIMO system in which the TX with $N_{T}$ antennas serves the RX with $N_{R}$ antennas. $N_{S}$ is the number of the data streams to be transmitted. The TX and the RX are equipped with $N_{T}^{RF}$ and $N_{R}^{RF}$ RF chains such that $N_{S}\leq N_{T}^{RF}\leq N_{T}$ and $N_{S}\leq N_{R}^{RF}\leq N_{R}$. This hybrid structure enables the TX to apply a baseband precoder $F_{b}\in C^{N_{T}^{RF}\times N_{S}}$ to the transmit signal $s\in C^{N_{S}}$ such that $E\left[ss^{H}\right]=I_{N_{S}}/N_{S}$, followed by an analog precoder $F_{a}\in C^{N_{T}\times N_{T}^{RF}}$. The TX has the power constraint as ${∥F_{a}F_{b}∥}_{F}^{2}=N_{S}$. Since the analog precoder $F_{a}$ is implemented using analog phase shifters, its elements have equal norm, i.e., ${\left[{\left[F_{a}\right]}_{:,i}{\left[F_{a}\right]}_{:,i}^{H}\right]}_{i,i}=N_{T}^{-1}$. Therefore, the transmitted signal is given by $x=F_{a}F_{b}s$.

We consider a narrowband block-fading channel and the received signal can be written as
(1)$$\tilde{y}=\sqrt{\rho }HF_{a}F_{b}s+n,$$
where $\tilde{y}\in C^{N_{R}\times 1}$, ρ is the average received power, $H\in C^{N_{R}\times N_{T}}$ is the channel matrix, $n\in C^{N_{R}\times 1}$ is the additive white Gaussian noise (AWGN) with $n∼\mathrm{CN}(0,\sigma^{2}I_{N_{R}})$. The mmWave channel H, which consists of $N_{c}$ clusters with $N_{ray}$ propagating rays [2], can be written as
(2)$$H=\gamma \sum_{i=1}^{N_{c}}\sum_{l=1}^{N_{ray}}\alpha_{il}\Lambda_{R}\left(\Theta_{R}^{\left(il\right)}\right)\Lambda_{T}\left(\Theta_{T}^{\left(il\right)}\right)a_{R}\left(\Theta_{R}^{\left(il\right)}\right)a_{T}^{H}\left(\Theta_{T}^{\left(il\right)}\right),$$
where $\gamma =\sqrt{N_{T}N_{R}/\left(N_{c}N_{ray}\right)}$ is a normalization factor. $\alpha_{il}$ is the complex channel gain of the *l*th propagation path in the *i*th scattering cluster. $\phi_{T}^{\left(il\right)}\left(\theta_{T}^{\left(il\right)}\right)$ and $\phi_{R}^{\left(il\right)}\left(\theta_{R}^{\left(il\right)}\right)$ represent the azimuth (elevation) angles of departure and arrival, respectively. $\Theta_{T}^{\left(il\right)}=\left(\phi_{T}^{\left(il\right)},\theta_{T}^{\left(il\right)}\right)$ and $\Theta_{R}^{\left(il\right)}=\left(\phi_{R}^{\left(il\right)},\theta_{R}^{\left(il\right)}\right)$ represent the angle of departure (AoD) and the angle of arrival (AoA), respectively. $\Lambda_{T}\left(\Theta_{T}^{\left(il\right)}\right)$ and $\Lambda_{R}\left(\Theta_{R}^{\left(il\right)}\right)$ denote the gains of transmit and receive antenna element that correspond to different AoDs and AoAs. $a_{T}\left(\Theta_{T}^{\left(il\right)}\right)\in C^{N_{T}}$ and $a_{R}\left(\Theta_{R}^{\left(il\right)}\right)\in C^{N_{R}}$ are the array response vectors at the TX and the RX. Considering the uniform planar array (UPA) with *U* elements on the *y*-axis and *V* elements on *z*-axis, the array response vector can be expressed as
(3)$$a\left(\phi ,\theta \right)=\frac{1}{\sqrt{N}}{\left[1,\ldots ,e^{j\frac{2\pi }{\lambda }d\left(u\sin\left(\phi \right)\sin\left(\theta \right)+v\cos\left(\theta \right)\right)},\ldots ,e^{j\frac{2\pi }{\lambda }d\left(\left(U-1\right)\sin\left(\phi \right)\sin\left(\theta \right)+\left(V-1\right)\cos\left(\theta \right)\right)}\right]}^{T},$$
where 0≤u<U, 0≤v<V and N=UV. λ denotes the wavelength of mmWave and *d* is the space between adjacent antennas.

The decoded data streams y, after being processed by analog and baseband combiners, can be written as
(4)$$\begin{array}{cc} y & =W_{b}^{H}W_{a}^{H}\tilde{y}\\ \; & =\sqrt{\rho }W_{b}^{H}W_{a}^{H}HF_{a}F_{b}s+W_{b}^{H}W_{a}^{H}n, \end{array}$$
where $W_{a}\in C^{N_{R}\times N_{R}^{RF}}$ and $W_{b}\in C^{N_{R}^{RF}\times N_{S}}$ denote the analog and baseband combiners, respectively. Similar to $F_{a}$, $W_{a}$ is subject to ${\left[{\left[W_{a}\right]}_{:,i}{\left[W_{a}\right]}_{:,i}^{H}\right]}_{i,i}=N_{R}^{-1}$.

Our objective is to design the hybrid precoders and combiners $F_{a}$, $F_{b}$, $W_{a}$, $W_{b}$ at the TX so as to maximize the spectral efficiency of the system. Assuming that the symbol vector s follows Gaussian distribution, the spectral efficiency can be written as
(5)$$R=\log_{2}\left(\left|I_{N_{S}}+\rho N_{S}^{-1}R_{n}^{-1}W_{b}^{H}W_{a}^{H}HF_{a}F_{b}F_{b}^{H}F_{a}^{H}H^{H}W_{a}W_{b}\right|\right),$$
where $R_{n}^{}=\sigma_{n}^{2}W_{b}^{H}W_{a}^{H}W_{a}W_{b}$ is the covariance matrix of the noise after receive processing.

It is necessary for the TX to obtain the instantaneous CSI for optimal precoders and combiners design. For simplicity, we assume that the perfect downlink CSI has been obtained at RX via pilot-based training and only focus on the design of joint CSI feedback and hybrid precoding.

To reduce feedback overhead, the RX first compresses H into a *M*-dimensional codeword c, then feeds back c (other than H) to the TX. This is described as
(6)c=F(H),
where $c\in C^{M\times 1}$ and $F\left(\cdot \right)$ represents the CSI compression scheme adopted at the RX.

The TX receives c and designs the downlink precoders and combiners accordingly. Note that, as illustrated in Figure 1, conventional hybrid precoding design schemes assume that the TX conducts design based on CSI feedback, i.e., CSI reconstruction from c is required, which may induce additional errors in this process. Differently, we design the precoders and combiners directly from c without requiring a CSI reconstruction process (the TX needs extra overhead to transmit the designed combiners to the RX). This is described as
(7)$$\left\{F_{a},F_{b},W_{a},W_{b}\right\}=P\left(c\right),$$
where $P\left(\cdot \right)$ denotes the hybrid precoding scheme.

In the following, we aim to jointly design $F\left(\cdot \right)$ in (Equation 6) and $P\left(\cdot \right)$ in (Equation 7), to maximize the spectral efficiency of the system (described in (Equation 5)). Such a problem is in general difficult to tackle with conventional optimization techniques. Alternatively, we seek a deep learning framework for handling this problem.

## 3. Proposed Deep Learning Framework for CSI Feedback and Hybrid Precoding

In this section, we describe the details of our proposd deep learning framework for the joint design of CSI feedback and hybrid precoding. Then, we describe how to generate the training dataset.

### 3.1. Deep Learning-Based Scheme

Figure 2 shows the architecture of our proposed neural network. It consists of a CSI feedback phase and a hybrid precoding phase. We build a CNN model to compress the channel matrices into the feedback codewords at the RX. At the TX, we build a DNN model to design the hybrid precoders and combiners from the feedback codewords. It is worth mentioning that in the deployment phase, the RX has stringent requirements on response latency and energy consumption, which is challenging for neural network design and provides a new research direction. We can use neural network quantization [33] to reduce model size and optimize the trade-off between accuracy and efficiency.

#### 3.1.1. CSI Feedback

As assumed, the RX has obtained perfect CSI and needs to compress the downlink channel matrix H into a *M*-dimensional codeword. This module is built with H as input, c as output, and with multiple-layer CNNs to realize the function $F\left(\cdot \right)$ in (Equation 6). The input of the proposed CNN framework, named X, are the real and imaginary parts of H, i.e., ${\left[X\right]}_{:,:,1}=ℜ\left\{H\right\}$ and ${\left[X\right]}_{:,:,2}=ℑ\left\{H\right\}$. The first and second layers are both convolutional layers with 64 filters to generate feature maps. The size of each filter is 2×2 and the stride is 1. Batch normalization is introduced to each convolutional layer. Following the second layer, we use a fully connected layer with 1024 units. The rectified linear unit (ReLU) is adopted at the first three layers, where ReLU(x)=max(x,0). Finally, a fully connected layer, whose size is M×1, is used to generate the codeword c.

#### 3.1.2. Hybrid Precoding

Under the assumption of an error-free feedback channel between the TX and the RX, the TX obtains the codewords c fed back from the RX and designs the hybrid precoders and combiners accordingly. We design a DNN model to realize the function $P\left(\cdot \right)$ in (Equation 7). The codeword c is the input vector and z is the output vector whose size is Q×1. Note that because $F_{a}$ and $W_{a}$ are analog precoder and combiner matrices, we only need to extract the angle information of the elements in $F_{a}$ and $W_{a}$. The vector z is the vectorized version of $F_{a},F_{b},W_{a},W_{b}$ and can be formed as
(8)$$z=[vec^{T}\left(∠F_{a}\right),vec^{T}\left(∠W_{a}\right),ℜ\left(vec^{T}\left(F_{b}\right)\right),ℑ\left(vec^{T}\left(F_{b}\right)\right),ℜ\left(vec^{T}\left(W_{b}\right)\right),ℑ\left(vec^{T}\left(W_{b}\right)\right)],$$
where $Q=N_{T}N_{T}^{RF}+N_{R}N_{R}^{RF}+2N_{T}^{RF}N_{S}+2N_{R}^{RF}N_{S}$. The first and second layers of the DNN are both fully connected layers with 1024 units. The activation function ReLU and the dropout layer with 50% probability are placed after the first and second layers. The third layer is a fully connected layers with *Q* units, which is used to generate the vector z.

### 3.2. Dataset Generation

The dataset of the proposed network is denoted as D, and a sample in D is an input-output pair written as $\left(X,z\right)$. In this paper, we need to design $F_{a},F_{b},W_{a},W_{b}$ from H to maximize the spectral efficiency. The optimization problem can be formulated as
(9)$$\begin{array}{cc} \underset{F_{a},F_{b},W_{a},W_{b}}{\mathrm{maximize}} & R\\ s.t. & F_{a}\in F_{a},\\ & W_{a}\in W_{a},\\ & \left|\right|F_{a}F_{b}{\left|\right|}_{F}^{2}=N_{S}, \end{array}$$
where $F_{a}$ and $W_{a}$ are the sets including all the feasible candidates of analog precoders and combiners, respectively. It is difficult to obtain the optimal solution of (Equation 9). To solve the problem and obtain the sub-optimal solution, $F_{a}$ and $W_{a}$ need to be predefined. Because the analog precoder $F_{a}$ is related to the array responses $a_{T}\left(\Theta_{T}\right)$ [2], $F_{a}$ can be defined as
(10)$$F_{a}=\left\{F_{a}^{\left(1\right)},\ldots ,F_{a}^{\left(c_{F}\right)},\ldots ,F_{a}^{\left(C_{F}\right)}\right\},$$
where $c_{F}=1,2,\ldots ,C_{F}$. $C_{F}$ = NpathNTRF is the number of the analog precoder candidates and $N_{path}=N_{c}\times N_{ray}$. $F_{a}^{\left(c_{F}\right)}=\left[a_{T}\left(\Theta_{T}^{\left(1\right)}\right),\ldots ,a_{T}\left(\Theta_{T}^{\left(t\right)}\right),\ldots ,a_{T}\left(\Theta_{T}^{\left(N_{T}^{RF}\right)}\right)\right]\in C^{N_{T}\times N_{T}^{RF}}$ is the candidate of $F_{a}$ in $F_{a}$, where $t=1,\ldots ,N_{T}^{RF}$. Similarly, the set of feasible analog combiners $W_{a}$ can be defined as
(11)$$W_{a}=\left\{W_{a}^{\left(1\right)},\ldots ,W_{a}^{\left(c_{W}\right)},\ldots ,W_{a}^{\left(C_{W}\right)}\right\},$$
where $c_{W}=1,2,\ldots ,C_{W}$. $C_{W}$ = NpathNRRF is the number of the analog combiner candidates. $W_{a}^{\left(c_{W}\right)}=[a_{R}\left(\Theta_{R}^{\left(1\right)}\right),\ldots ,a_{R}\left(\Theta_{R}^{\left(p\right)}\right),$$\ldots ,a_{R}\left(\Theta_{R}^{\left(N_{R}^{RF}\right)}\right)]\in C^{N_{R}\times N_{R}^{RF}}$ is the candidate of $W_{a}$ in $W_{a}$, where $p=1,\ldots ,N_{R}^{RF}$. Therefore, the optimization problem in (Equation 9) can be rewritten as
(12)$$\begin{array}{cc} \underset{{\hat{c}}_{F},{\hat{c}}_{W}}{\mathrm{maximize}} & R\\ s.t. & F_{a}\in F_{a},\\ & W_{a}\in W_{a},\\ & F_{b}={\left(F_{a}^{H}F_{a}\right)}^{-1}F_{a}^{H}F^{opt},\\ & W_{b}={\left(W_{a}^{H}AW_{a}\right)}^{-1}\left(W_{a}^{H}AW^{opt}\right), \end{array}$$
where $A=\frac{\rho }{N_{S}}HF_{a}F_{b}F_{b}^{H}F_{a}^{H}H^{H}+\sigma_{n}^{2}I_{N_{R}}$ is the covariance of the array output in (Equation 1). $F^{opt}$, $W^{opt}$ represent the optimal fully-digital precoder and combiner that can be obtained from singular value decomposition (SVD) of H [2,23].

To reduce the complexity, the problem (Equation 12) can be decomposed into the sub-problems of precoder and combiner designs. The precoder design problem (Equation 13) and combiner design problem (Equation 14) can be written as [23]
(13)$$\begin{array}{cc} \underset{{\hat{c}}_{F}}{\mathrm{maximize}} & \log_{2}\left(\left|I_{N_{S}}+\frac{\rho }{N_{S}\sigma_{n}^{2}}{\left(W^{opt^{H}}W^{opt}\right)}^{-1}W^{opt^{H}}HF_{a}F_{b}F_{b}^{H}F_{a}^{H}H^{H}W^{opt}\right|\right)\\ s.t. & F_{a}\in F_{a},\\ & F_{b}={\left(F_{a}^{H}F_{a}\right)}^{-1}F_{a}^{H}F^{opt}, \end{array}$$
(14)$$\begin{array}{cc} \underset{{\hat{c}}_{W}}{\mathrm{maximize}} & \log_{2}\left(\left|I_{N_{S}}+\frac{\rho }{N_{S}\sigma_{n}^{2}}{\left(W_{b}^{H}W_{a}^{H}W_{a}W_{b}\right)}^{-1}W_{b}^{H}W_{a}^{H}HF^{opt}F^{opt^{H}}H^{H}W_{a}W_{b}\right|\right)\\ s.t. & W_{a}\in W_{a},\\ & W_{b}={\left(W_{a}^{H}AW_{a}\right)}^{-1}\left(W_{a}^{H}AW^{opt}\right). \end{array}$$

In this case, the Euclidean distance between the optimal fully-digital precoder (combiner) and the hybrid precoders (combiners) is minimized, which will maximize the spectral efficiency of hybrid precoding. Once we solve (Equation 13) and (Equation 14) and obtain ${\hat{c}}_{F}$, ${\hat{c}}_{W}$, $F_{a},F_{b},W_{a},W_{b}$ can be constructed and the dataset D can be generated.

## 4. Implementation Details

In data generation, we generate the channel matrix H according to (Equation 2). We consider the UPA with $N_{T}=36$ and $N_{R}=36$ for the TX and RX, respectively. The number of the RF chains at the TX and the RX are both set as 4, i.e., $N_{T}^{RF}=N_{R}^{RF}=4$. For each channel matrix, the propagation environment is modeled with $N_{c}=4$ and $N_{ray}=4$ for each clusters with $\sigma_{\Theta }^{2}=5^{\circ }$ for all transmit and receive azimuth and elevation angles, which are uniform and randomly selected from the interval $\left[-60^{\circ },60^{\circ }\right]$ and $\left[-30^{\circ },30^{\circ }\right]$, respectively. The frequency is set as 28 GHz, and the antenna spacing is half the wavelength.

We implement the proposed neural network using MATLAB as a simulation environment. Notably, the channel matrices have been normalized before inputting the neural network. The typical mean squared error (MSE) between the label z and the actual output $\hat{z}$ is computed as the loss function, which is described as
(15)$$Loss=\frac{1}{J}\sum_{j=1}^{J}{∥z_{j}-{\hat{z}}_{j}∥}_{2}^{2},$$
where *J* denotes the size of the dataset. Stochastic gradient descent with momentum (SGDM) optimizer is used to reduce the loss and update the weight of the network. The epoch, batch size, and initial learning rate are set as 200, 400, and 0.0005, respectively. The learning rate is decreased after 20 epochs by a factor of 0.9.

## 5. Experiment Results

In this section, we evaluate the spectral efficiency of the proposed neural network and compare the performance with the following benchmarks:

**Benchmark 1: SVD with perfect CSI [2]**: Considering that the TX has obtained the perfect CSI, the TX performs hybrid precoding using a fully-digital precoder and combiner, which can be obtained from the SVD of the channel matrix H. In this case, the upper bound of spectral efficiency can be obtained.

**Benchmark 2: MO-AltMin with perfect CSI** [6]: Given the perfect CSI at the TX, the TX performs hybrid precoding by using the MO-AltMin algorithm. The MO-AltMin algorithm is one of the alternate minimization hybrid precoding schemes. It is based on manifold optimization and has the best performance in [6].

**Benchmark 3: SOMP with perfect CSI** [2]: Given the perfect CSI at the TX, the SOMP, which is a greedy-based algorithm, is used by the TX to design the hybrid precoders and combiners.

**Benchmark 4: MO-AltMin with CsiNet** [11]: In this benchmark, no prior perfect CSI is initially assumed at the TX. The RX needs to feed the CSI back to the TX over finite-capacity links. To compare the performance of our proposed scheme that the TX designs the precoders and combiners from the codewords directly, and conventional schemes that the TX designs the precoders and combiners from the channel matrices reconstructed from the codewords, we implement a scheme that uses a deep learning approach to perform channel feedback and reconstruction, followed by a conventional hybrid precoding algorithm. We choose CsiNet, which is a classical CSI sensing and recovery mechanism, to realize the channel feedback and reconstruction. The TX uses MO-AltMin algorithm to design the hybrid precoders and combiners after reconstructing the channel matrices.

Figure 3 presents the spectral efficiency comparison of different schemes versus SNRs. The number of data streams $N_{S}$ is set as 2 and the length of codewords *M* is set as 25. It can be observed that the spectral efficiency of all considered algorithms increases monotonically with increasing SNR. The SVD with perfect CSI has the best performance. We observe that our proposed method can approach the performance of the MO-AltMin with perfect CSI, which means that our end-to-end neural network can effectively generate the precoders and combiners, which maximizes the spectral efficiency. In addition, our proposed scheme also has better performance than the MO-AltMin with CsiNet in the same codeword length, which verifies that our proposed end-to-end method can get better performance than the traditional separate design method in this situation. The SOMP with perfect CSI has the worst performance because it cannot select the optimal set of array responses from the dictionary.

We further investigate the performance of our proposed method and conventional hybrid precoding design approaches versus the length of codewords. In Figure 4, as the length of codewords increases, the spectral efficiency of our proposed scheme can gradually approach and eventually exceed the MO-AltMin with perfect CSI when M=30. Note that the MO-AltMin with perfect CSI suffers from very high feedback overhead, which means that our proposed scheme has lower feedback overhead with similar performance. In addition, it can be observed that our proposed scheme outperforms the MO-AltMin with CsiNet in the same codeword length and the gap is significantly large when *M* is small, e.g., M=5. This observation indicates the superior performance of the proposed end-to-end neural network approach for FDD mmWave massive MIMO systems in the case of very low CSI feedback overhead.

Finally, we compare the computational complexity for our proposed method and different benchmarks. Table 1 shows that our proposed method has much lower running time than other benchmarks. It means that our proposed method can be executed with a relatively lower overhead and is more suitable for practical scenarios.

## 6. Conclusions

In this paper, we consider the joint design of CSI feedback and hybrid precoding for FDD massive MIMO systems. We propose a new deep learning-based end-to-end method that bypasses channel reconstruction and directly designs the hybrid precoders and combiners from the feedback codewords for FDD massive MIMO systems. We propose a new neural network that jointly optimizes CSI feedback and hybrid precoding. In order to train the network, we generate the input-output pairs, where the input is the channel matrices and the output is the hybrid precoders and combiners. Numerical results indicate the ability of the proposed network in reducing the feedback overhead and boosting the system performance in terms of spectral efficiency, especially in the case of the limited feedback resources. Future research directions include some other transmission modules, e.g., the downlink pilot transmission and quantized CSI feedback. Moreover, the performance of our proposed method in terms of energy efficiency is another promising direction for future works.

## Figures and Tables

**Figure 1 entropy-24-00441-f001:**
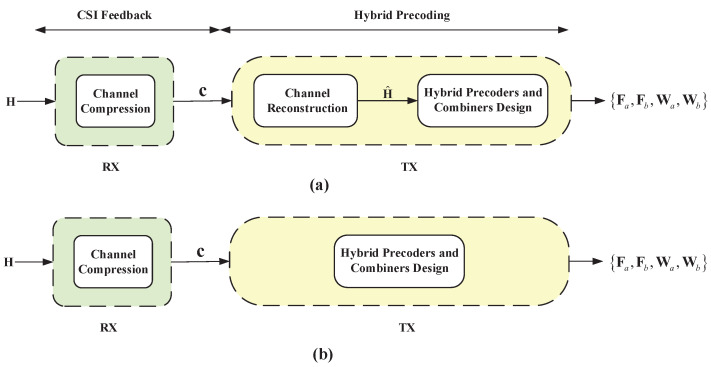
(**a**) The architecture of the traditional hybrid precoding methods that reserve channel reconstruction. (**b**) Our proposed hybrid precoding method that bypasses channel reconstruction (assuming that the RX has obtained the perfect CSI through the pilot training in both architectures).

**Figure 2 entropy-24-00441-f002:**
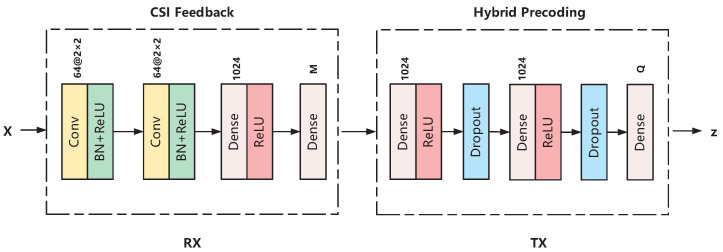
The architecture of the proposed neural network that represents the end-to-end CSI feedback and hybrid precoding.

**Figure 3 entropy-24-00441-f003:**
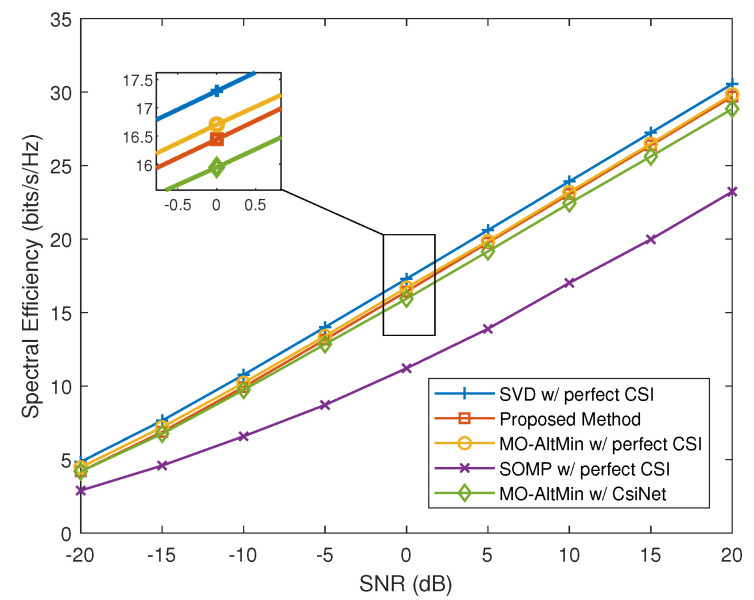
Spectral efficiency versus SNRs for $N_{R}=N_{T}=36$, $N_{S}=2$, M=25.

**Figure 4 entropy-24-00441-f004:**
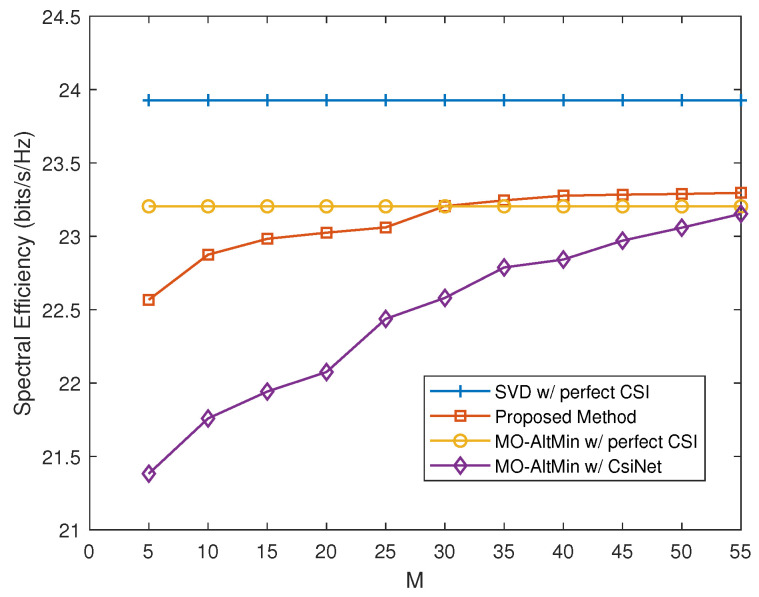
Spectral efficiency versus the length of codewords *M*.

**Table 1 entropy-24-00441-t001:** Comparison of Computational Complexity.

Methods	Running Time
Proposed method	0.0046 s
MO-AltMin with perfect CSI	1.2999 s
MO-AltMin with CsiNet	1.3007 s

## Data Availability

Not applicable.

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
