# Peer review of "Deep Learning-Based Joint CSI Feedback and Hybrid Precoding in FDD mmWave Massive MIMO Systems"

_entropy, 2022, doi:10.3390/e24040441_

Round 1

Reviewer 1 Report

The authors present a deep learning-based approach for a point-to-point FDD
mmWave massive MIMO system and the numerical results prove the efficiency of the technique. The quality of the submitted work is high and it is suggested for publication.

Reviewer 2 Report

Deep Learning-Based Joint CSI Feedback and Hybrid Precoding in FDD mmWave Massive MIMO Systems

  1. This paper presents a new end-to-end learning method bypassing the channel reconstruction phase, and design the hybrid precoders and combiners directly from the feedback code words (a trodden version of the CSI). More explicitly, the authors enterprise a neural network composed of the CSI feedback and hybrid precoding. Simulation results show that the proposed network can achieve better performance than conventional hybrid precoding schemes which reserve channel reconstruction, especially when the feedback resources are limited. The purpose of this study is to identify the ability of the proposed network in reducing the feedback overhead and boosting the system performance in terms of spectral efficiency, especially in the case of the limited feedback resources. The paper quality is good, its idea is very clear. However, the quality of the paper should be further improved by carefully addressing the following issues.
  2. The main contribution should be more highlighted to distinguish it from the recent existing works. Furthermore, you should cite more recent works in the introduction section as related works. However, we can find several new works (published in 2020-2021) that study the same work research. Such works are not cited in the paper.
  3. The authors should compare their work with more recent existing works that focused on the same research problem with other methods and not just with their earlier research.

  1. The authors are suggested to interpret the challenges introduced by the deployment of the proposed method. The authors should focus on the mutual challenging issues and growing research instructions, by studying the balance between obtained accuracy, complexity and efficiency.
  2. More interpretation and justification are needed in the Methods, Example of application, and evaluation sections. You should show the big difference between the different compared algorithms. Furthermore, you should more present and highlight the key innovation that you have proposed to enhance the Maximum Entropy Method.

  1. The Performance Analysis can be more enhanced by more comparisons between your work and other recent work results in the form of numerical or plot presentations. Furthermore, it is preferable to study the performance of your proposed approach in terms of energy efficiency and system complexity. And also, you can use other performance metrics.
  2. You need to add more evaluation to prove that your method outperforms the other methods.
  3. The obtained results (in Figures and tables) are not well commented on; however, the Evaluation section is not well organized and does not describe very well the obtained results. You should add more interpretation and justification. Also, the paper organization needs to be reviewed.
  4. The conclusion section does not summarize very well the main points given in the work. Generally, it should focus on the main contributions and results of the work. However, it can also include a description of limitations, suggest some further research, and give a concluding statement.
  5. The writing quality needs to be more improved by avoiding typos and editing errors. However, some sentences are not well structured and it does not very well describe the ideas, and the meaning of other sentences is not well clear. And, there are many missed or added ‘s’ and ‘the’. Please, check the language carefully many times.
  6. Add this references in the list of the references “González-Coma, J.P.; Suárez-Casal, P.; Castro, P.M.; Castedo, L. FDD Channel Estimation Via Covariance Estimation in Wideband Massive MIMO Systems. Sensors2020, 20, 930. https://doi.org/10.3390/s20030930”
